# MOGONET integrates multi-omics data using graph convolutional networks allowing patient classification and biomarker identification

Tongxin Wang [1,8], Wei Shao [2,8], Zhi Huang[2,3], Haixu Tang[1], Jie Zhang [4], Zhengming Ding [5 ✉] & Kun Huang [2,6,7 ✉]

To fully utilize the advances in omics technologies and achieve a more comprehensive understanding of human diseases, novel computational methods are required for integrative analysis of multiple types of omics data. Here, we present a novel multi-omics integrative method named Multi-Omics Graph cOnvolutional NETworks (MOGONET) for biomedical classification. MOGONET jointly explores omics-specific learning and cross-omics correlation learning for effective multi-omics data classification. We demonstrate that MOGONET outperforms other state-of-the-art supervised multi-omics integrative analysis approaches from different biomedical classification applications using mRNA expression data, DNA methylation data, and microRNA expression data. Furthermore, MOGONET can identify important biomarkers from different omics data types related to the investigated biomedical problems.

[1] Department of Computer Science, Indiana University Bloomington, Bloomington, IN, USA. [2] Department of Medicine, Indiana University School of Medicine, Indianapolis, IN, USA. [3] School of Electrical and Computer Engineering, Purdue University, West Lafayette, IN, USA. [4] Department of Medical and Molecular Genetics, Indiana University School of Medicine, Indianapolis, IN, USA. [5] Department of Computer Science, Tulane University, New Orleans, LA, USA. [6] Department of Biostatistics and Health Data Science, Indiana University School of Medicine, Indianapolis, IN, USA. [7] Regenstrief Institute, Indianapolis, IN, USA. [8] These authors contributed equally: Tongxin Wang, Wei Shao. ✉email: zding1@tulane.edu; kunhuang@iu.edu

The rapid advancement in high-throughput biomedical technologies has enabled the collection of various types of "omics" data with unprecedented details. Genome-wide data for different molecular processes, such as mRNA expression, DNA methylation, and microRNA (miRNA) expression, can be acquired for the same set of samples, resulting in multiple omics (multi-omics) data for various studies of diseases. While each omics technology can only capture part of the biological complexity, integrating multiple types of omics data can provide a more holistic view of the underlying biological processes. Specifically, for human diseases, existing studies have demonstrated that integrating data from multiple omics technologies can improve the accuracy of patient clinical outcome prediction comparing with only using a single type of omics data[1–7]. Therefore, there is a need for novel integrative analysis methods to effectively take advantage of the interactions and complementary information in multi-omics data.

A great number of methods have been proposed over the years to perform multi-omics data integration for various problems. However, most existing efforts focus on unsupervised multi-omics data integration without the additional information of sample labels[8–10]. With the rapid development of personalized medicine, curated datasets with detailed annotations that characterize the phenotypes or traits of the samples are becoming more widely available. Therefore, there is an increasing interest in supervised multi-omics integration methods that can identify disease-related biomarkers and perform predictions on new samples. Early attempts of supervised data integration methods for biomedical classification tasks include feature concatenation-based strategies and ensemble-based strategies. On the one hand, concatenation-based methods integrated different omics data types by directly concatenating the input data features to learn the classification model[5]. On the other hand, ensemble-based methods integrated the predictions from different classifiers, each trained on one type of omics data individually[1]. However, these methods failed to consider the correlations among different omics data types and could be biased toward certain omics data types. Recently, more supervised multi-omics data integration methods focused on exploiting the interactions across different omics data types have been proposed. For example, van de Wiel et al.[6] introduced an adaptive group-regularized ridge regression method that incorporated methylation microarray data and curated annotations of methylation probes for cervical cancer diagnostic classification. Singh et al.[4] proposed Data Integration Analysis for Biomarker discovery using Latent cOmponents (DIABLO) by extending the sparse generalized canonical correlation analysis to a supervised setting, which could seek common information across multiple omics types while discriminating between different phenotypic groups.

With the continuous advancement of deep learning in various tasks, more and more multi-omics integration methods begin to take advantage of the high learning capability and flexibility of deep neural networks (NN)[2,11–13]. For example, Huang et al.[2] integrated the features of mRNA expression and miRNA expression data with additional clinical information at hidden layers for better prognosis prediction in breast cancer. However, these existing methods are based on fully connected networks, which did not exploit the correlations between samples effectively through similarity networks. Moreover, while current deep learning-based methods often integrate different omics data at the input space[11,13] or the learned feature space[2,12], different types of omics data can also present unique characteristics at the high-level label space. Therefore, it is crucial to utilize the correlations across different classes and different omics data types to further boost the learning performance.

To this end, we introduce MOGONET, a multi-omics data analysis framework for classification tasks in biomedical applications. MOGONET unifies omics-specific learning with multi-omics integrative classification at the label space. Specifically, MOGONET utilizes graph convolutional networks (GCN) for omics-specific learning. Comparing with the fully connected NN, GCN takes advantage of both the omics features and the correlations among samples described by the similarity networks for better classification performance. Besides directly concatenating the label distribution from each omics data type, MOGONET also utilizes View Correlation Discovery Network (VCDN) to explore the cross-omics correlations at the label space for effective multi-omics integration. To the best of our knowledge, MOGONET is the first supervised multi-omics integrative method that utilizes GCNs for omics data learning to perform effective class prediction on new samples. We demonstrated the capabilities and versatility of MOGONET through a wide range of biomedical classification applications, including Alzheimer's disease patient classification, tumor grade classification in low-grade glioma (LGG), kidney cancer type classification, and breast invasive carcinoma subtype classification. We also showed the necessity of integrating multiple omics data types as well as the importance of combining both GCN and VCDN for multi-omics data classification through comprehensive ablation studies. Moreover, we demonstrated that MOGONET can identify important omics signatures and biomarkers related to the investigated biomedical problems.

## Results

**Framework of MOGONET.** We introduce MOGONET, a supervised multi-omics integration framework for biomedical classification tasks (Fig. 1). After preprocessing and feature pre-selection to remove noise and redundant features, we first use GCNs to learn the classification task with each omics data type individually. Specifically, we construct a weighted sample similarity network for each type of omics data using cosine similarity. Taking the input of both the omics features and the corresponding similarity network, a GCN is trained for each omics data type to generate initial predictions of class labels. A major advantage of GCNs is that they can exploit the information from both the omics data and the correlations between samples for better prediction. Then, initial predictions generated by each omics-specific GCN are further utilized to construct the cross-omics discovery tensor, which reflects the cross-omics label correlations. Finally, the cross-omics discovery tensor is reshaped into a vector and forwarded to VCDN for final label prediction. VCDN can effectively integrate initial predictions from each omics-specific network by exploring the latent correlations across different omics data types in the higher-level label space. MOGONET is an end-to-end model, and omics-specific GCNs and VCDN are trained alternatively until convergence. To this end, the final prediction from MOGONET is based on both effective omics-specific predictions generated by GCNs and the learned cross-omics label correlation knowledge generated by VCDN. To the best of our knowledge, MOGONET is the first method to explore both GCNs and cross-omics relationships in the label space for effective multi-omics integration in biomedical data classification tasks.

**Datasets.** To demonstrate the effectiveness of MOGONET, we applied the proposed method on four different biomedical classification tasks using four different datasets: ROSMAP for Alzheimer's Disease (AD) patients vs. normal control (NC) classification, LGG for grade classification in low-grade glioma (LGG), KIPAN for kidney cancer type classification, and BRCA for breast invasive carcinoma (BRCA) PAM50 subtype classification. Three types of omics data (i.e., mRNA expression data

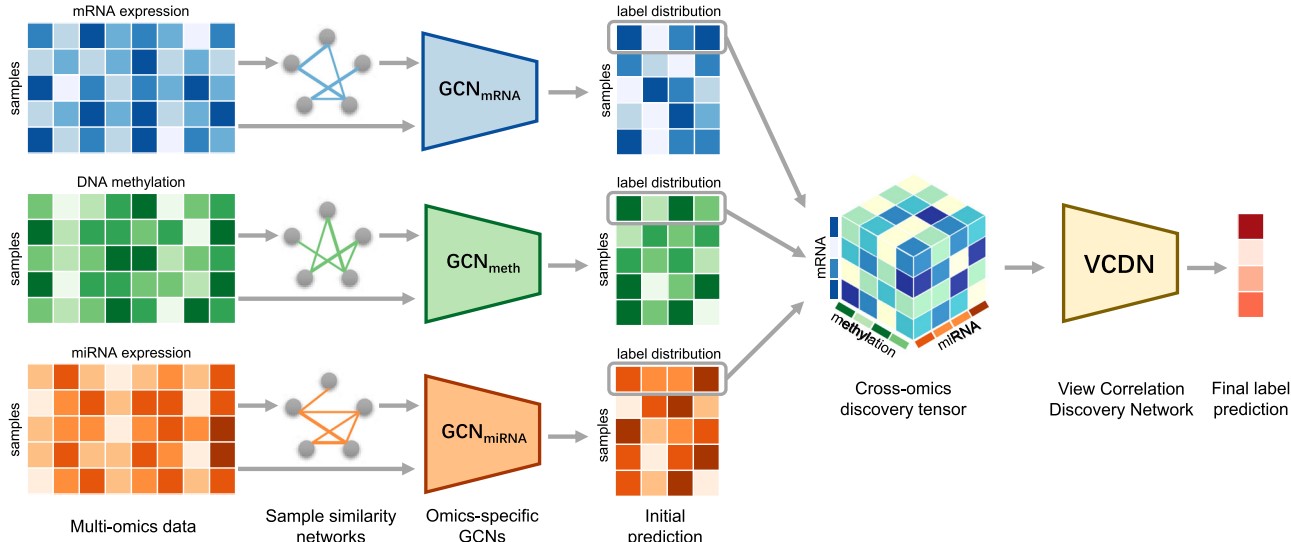

**Fig. 1 Illustration of MOGONET.** MOGONET combines GCN for multi-omics-specific learning and VCDN for multi-omics integration. For clear and concise illustration, an example of one sample is chosen to demonstrate the VCDN component for multi-omics integration. Preprocessing is first performed on each omics data type to remove noise and redundant features. Each omics-specific GCN is trained to perform class prediction using omics features and the corresponding sample similarity network generated from the omics data. The cross-omics discovery tensor is calculated from the initial predictions of omics-specific GCNs and forwarded to VCDN for final prediction. MOGONET is an end-to-end model and all networks are trained jointly.

| Table 1 Summary of datasets. | | | |
|---|---|---|---|
| **Dataset** | **Categories** | **Number of features mRNA, meth, miRNA** | **Number of features for training mRNA, meth, miRNA** |
| ROSMAP | NC: 169, AD: 182 | 55,889, 23,788, 309 | 200, 200, 200 |
| LGG | Grade 2: 246, Grade 3: 264 | 20,531, 20,114, 548 | 2000, 2000, 548 |
| KIPAN | KICH: 66, KIRC: 318, KIRP: 274 | 20,531, 20,111, 445 | 2000, 2000, 445 |
| BRCA | Normal-like: 115, Basal-like: 131, HER2-enriched: 46, Luminal A: 436, Luminal B: 147 | 20,531, 20,106, 503 | 1000, 1000, 503 |

mRNA refers to mRNA expression data. meth refers to DNA methylation data. miRNA refers to miRNA expression data. The ROSMAP dataset is for the classification of Alzheimer's disease (AD) patients vs. normal control (NC). The LGG dataset is for grade classification in low-grade glioma (LGG). The KIPAN dataset is for kidney cancer type classification with chromophobe renal cell carcinoma (KICH), clear renal cell carcinoma (KIRC), and papillary renal cell carcinoma (KIRP). The BRCA dataset is for breast invasive carcinoma (BRCA) PAM50 subtype classification with normal-like, basal-like, human epidermal growth factor receptor 2 (HER2)-enriched, Luminal A, and Luminal B subtypes.

(mRNA), DNA methylation data (meth), and miRNA expression data (miRNA)) were used for classification to provide comprehensive and complementary information on the diseases. Only samples with matched mRNA expression, DNA methylation, and miRNA expression data were included in our study. The details of the datasets are listed in Table 1. Since noise redundant features may affect the performance of classification tasks, preprocessing and feature preselection were performed on each omics data type individually, and the number of features used for training was also listed in Table 1.

Omics data of LGG, KIPAN, and BRCA, as well as the grade information of LGG patients, were acquired from The Cancer Genome Atlas Program (TCGA) through Broad GDAC Firehose. PAM50 is a 50-gene signature that classifies breast cancer into five molecular subtypes: normal-like, basal-like, human epidermal growth factor receptor 2 (HER2)-enriched, Luminal A, and Luminal B[14,15]. The PAM50 breast cancer subtype information of TCGA BRCA patients was acquired through TCGAbiolinks[16]. Different omics data in the ROSMAP dataset were acquired from AMP-AD Knowledge Portal[17]. ROSMAP is composed of ROS and MAP, which both are longitudinal clinical-pathologic cohort studies of AD from Rush University[18,19].

For kidney cancer type classification, the differences among chromophobe renal cell carcinoma (KICH), clear renal cell carcinoma (KIRC), and papillary renal cell carcinoma (KIRP) can be clearly observed in the omics data. Therefore, kidney cancer type classification is the simplest task among these classification tasks and serves more as a proof-of-concept experiment for multi-class applications. On the other hand, while multi-omics integration methods have been well studied for cancers[2,4,11–13], analysis of AD with multiple omics data types is an emerging field. For example, Jiang et al.[20] analyzed mRNA and miRNA expression data to identify active transcription factors and miRNA regulatory pathways in AD to better understand the pathology of AD. Humphries et al.[21] combined RNA sequencing and DNA methylation data to identify gene networks specific to late-onset AD. However, methods that directly address the accurate identification of AD patients from normal age-matched people with machine learning algorithms are still limited. Here, to demonstrate the generalization ability of MOGONET to different diseases and medical applications, we also applied MOGONET on AD patient classification using the ROSMAP dataset, where AD patients and NC subjects were selected for the classification task in our experiment.

**Multi-omics classification performance evaluation.** We compared the classification performance of MOGONET with existing supervised multi-omics integration algorithms. We also

**Table 2 Classification results on ROSMAP dataset.**

| Method | ACC | F1 | AUC |
|---|---|---|---|
| KNN | 0.657 ± 0.036 | 0.671 ± 0.044 | 0.709 ± 0.045 |
| SVM | 0.770 ± 0.024 | 0.778 ± 0.016 | 0.770 ± 0.026 |
| Lasso | 0.694 ± 0.037 | 0.730 ± 0.033 | 0.770 ± 0.035 |
| RF | 0.726 ± 0.029 | 0.734 ± 0.021 | 0.811 ± 0.019 |
| XGBoost | 0.760 ± 0.046 | 0.772 ± 0.045 | 0.837 ± 0.030 |
| NN | 0.755 ± 0.021 | 0.764 ± 0.021 | 0.827 ± 0.025 |
| GRridge | 0.760 ± 0.034 | 0.769 ± 0.029 | 0.841 ± 0.023 |
| block PLSDA | 0.742 ± 0.024 | 0.755 ± 0.023 | 0.830 ± 0.025 |
| block sPLSDA | 0.753 ± 0.033 | 0.764 ± 0.035 | 0.838 ± 0.021 |
| NN_NN | 0.766 ± 0.023 | 0.777 ± 0.019 | 0.819 ± 0.017 |
| NN_VCDN | 0.775 ± 0.026 | 0.790 ± 0.018 | 0.843 ± 0.021 |
| MOGONET_NN (Ours) | 0.804 ± 0.016 | 0.808 ± 0.010 | 0.858 ± 0.024 |
| MOGONET (Ours) | 0.815 ± 0.023 | 0.821 ± 0.022 | 0.874 ± 0.012 |

**Table 3 Classification results on LGG dataset.**

| Method | ACC | F1 | AUC |
|---|---|---|---|
| KNN | 0.729 ± 0.034 | 0.738 ± 0.033 | 0.799 ± 0.038 |
| SVM | 0.754 ± 0.046 | 0.757 ± 0.050 | 0.754 ± 0.046 |
| Lasso | 0.761 ± 0.018 | 0.767 ± 0.022 | 0.823 ± 0.027 |
| RF | 0.748 ± 0.012 | 0.742 ± 0.010 | 0.823 ± 0.010 |
| XGBoost | 0.756 ± 0.040 | 0.767 ± 0.032 | 0.840 ± 0.023 |
| NN | 0.737 ± 0.023 | 0.748 ± 0.024 | 0.810 ± 0.037 |
| GRridge | 0.746 ± 0.038 | 0.756 ± 0.036 | 0.826 ± 0.044 |
| block PLSDA | 0.759 ± 0.025 | 0.738 ± 0.031 | 0.825 ± 0.023 |
| block sPLSDA | 0.685 ± 0.027 | 0.662 ± 0.030 | 0.730 ± 0.026 |
| NN_NN | 0.740 ± 0.039 | 0.756 ± 0.036 | 0.824 ± 0.036 |
| NN_VCDN | 0.740 ± 0.030 | 0.771 ± 0.021 | 0.826 ± 0.031 |
| MOGONET_NN (Ours) | 0.804 ± 0.025 | 0.811 ± 0.023 | 0.832 ± 0.029 |
| MOGONET (Ours) | 0.816 ± 0.016 | 0.814 ± 0.014 | 0.840 ± 0.027 |

performed comprehensive ablation studies to demonstrate the necessity of different components in MOGONET. To compare the effectiveness of different multi-omics integration methods, we randomly selected 30% of the samples in a dataset as the test set and the remaining 70% of the samples as the training set. The test set was constructed by preserving the class distribution in the original dataset. To evaluate the performance of the compared methods, we used accuracy (ACC), F1 score (F1), and area under the receiver operating characteristic curve (AUC) for binary classification tasks, and we used accuracy (ACC), average F1 score weighted by support (F1_weighted), and macro-averaged F1 score (F1_macro) for multi-class classification tasks. We evaluated all the methods on five different randomly generated training and testing splits, and the mean and standard deviation of the evaluation metrics across these five experiments were reported.

**MOGONET outperformed existing supervised multi-omics integration methods in various classification tasks.** We compared the classification performance of MOGONET with the following nine existing classification algorithms for omics data: (1) K-nearest neighbor classifier (KNN). Label predictions were made by voting of KNN in the training data. (2) Support vector machine classifier (SVM). (3) Linear regression trained with L1 regularization (Lasso). In Lasso, an individual model was trained to predict the probability of each class, and the class predicted with the highest probability was selected as the final prediction of the class label for the entire model. (4) Random forest classifier (RF). (5) Gradient tree boosting-based classifier implemented in the XGBoost package (XGBoost). (6) Fully connected NN

classifier. Deep fully connected NN were trained with cross-entropy loss. (7) Adaptive group-regularized ridge regression (GRridge)[6]. The implementation in the GRridge R package was used. (8) Block PLSDA. Block PLSDA is a multi-omics integration method that projects data to latent structures with discriminant analysis. Block PLSDA integrates multiple types of omics data measured on the same set of samples to classify a discrete outcome. Block PLSDA is one of the supervised analysis methods included in DIABLO[4]. (9) Block sPLSDA. Block sPLSDA is block PLSDA with additional sparse regularization, which can select relevant features from the dataset. It is also a supervised analysis method within DIABLO. Implementations in the mixOmics R package[22] were used for block PLSDA and block sPLSDA. Block PLSDA and block sPLSDA represent the state-of-the-art approaches for supervised multi-omics integration and classification. Among the tested methods, KNN, SVM, Lasso, RF, XGBoost, and NN were trained with the direct concatenation of the preprocessed multi-omics data as input. All methods were trained with the same preprocessed data. The classification results for ROSMAP, LGG, BRCA, and KIPAN are shown in Tables 2–4 and Supplementary Table 1, respectively.

From Tables 2–4 and Supplementary Table 1, we observed that MOGONET outperformed the compared multi-omics integration methods in most classification tasks. The only exception was in LGG grade classification, where XGBoost and MOGONET yielded the same average AUC. However, MOGONET still achieved better performance in LGG grade classification than XGBoost when evaluated with ACC and F1. Additionally, we evaluated the performance of multi-class classification tasks using average AUC score weighted by support (AUC_weighted) and

**Table 4 Classification results on BRCA dataset.**

| Method | ACC | F1_weighted | F1_macro |
|---|---|---|---|
| KNN | 0.742 ± 0.024 | 0.730 ± 0.023 | 0.682 ± 0.025 |
| SVM | 0.729 ± 0.018 | 0.702 ± 0.015 | 0.640 ± 0.017 |
| Lasso | 0.732 ± 0.012 | 0.698 ± 0.015 | 0.642 ± 0.026 |
| RF | 0.754 ± 0.009 | 0.733 ± 0.010 | 0.649 ± 0.013 |
| XGBoost | 0.781 ± 0.008 | 0.764 ± 0.010 | 0.701 ± 0.017 |
| NN | 0.754 ± 0.028 | 0.740 ± 0.034 | 0.668 ± 0.047 |
| GRridge | 0.745 ± 0.016 | 0.726 ± 0.019 | 0.656 ± 0.025 |
| block PLSDA | 0.642 ± 0.009 | 0.534 ± 0.014 | 0.369 ± 0.017 |
| block sPLSDA | 0.639 ± 0.008 | 0.522 ± 0.016 | 0.351 ± 0.022 |
| NN_NN | 0.796 ± 0.012 | 0.784 ± 0.014 | 0.723 ± 0.018 |
| NN_VCDN | 0.792 ± 0.010 | 0.781 ± 0.006 | 0.721 ± 0.018 |
| MOGONET_NN (Ours) | 0.805 ± 0.017 | 0.782 ± 0.030 | 0.737 ± 0.038 |
| MOGONET (Ours) | 0.829 ± 0.018 | 0.825 ± 0.016 | 0.774 ± 0.017 |

macro-averaged AUC score (AUC_macro), with the results for BRCA and KIPAN shown in Supplementary Tables 6 and 7. MOGONET achieved the best performance on the BRCA dataset when evaluated using the AUC metrics while achieved the same average AUC scores as GRridge on the KIPAN dataset. Note that the classification of kidney cancer type using the KIPAN dataset was a relatively simpler task served as a proof-of-concept experiment for multi-class applications, where all the compared methods achieved quite high performance in different metrics. However, MOGONET still outperformed GRridge when evaluated using ACC, F1_weighted, and F1_macro on the KIPAN dataset. Moreover, MOGONET consistently outperformed the state-of-the-art supervised multi-omics integration methods (i.e., block PLSDA and block sPLSDA) in different classification tasks, demonstrating the superiority of multi-omics data classification capability by combining GCNs for omics-specific learning and VCDN for multi-omics integration. Comparing with existing methods, the advantages of MOGONET were further demonstrated in difficult applications such as AD patient classification and BRCA subtype classification, indicating the superior learning capability of MOGONET. Interestingly, although deep learning-based methods have shown great promises in classification applications, the deep learning-based method NN did not show clear improvements over other approaches. This observation suggested that proper design of deep learning algorithms specific to supervised multi-omics integration applications was required to achieve superior classification performance.

Different subtypes of BRCA using PAM50 classification may indicate different tumorigenesis mechanisms, and some subtypes may be closer in terms of molecular features than the others. Therefore, we further evaluated the performance of different methods on BRCA PAM50 subtype classification with the following two additional definitions of labels while trained with the labels of five subtypes. One was the binary classification of normal-like vs. non-normal-like subtypes, where the non-normal-like category included the rest four different subtypes (Supplementary Table 2). The other one included four categories, where the Luminal A and Luminal B subtypes were merged into one category as they are more associated with each other than the rest of the subtypes (Supplementary Table 3)[14,15]. Note that for the same method, results in Table 4 and Supplementary Tables 2 and 3 were from the evaluation on the same set of models that performs prediction of five BRCA subtypes, while only different definitions of the labels were used for evaluation. When considered jointly, Table 4 and Supplementary Tables 2 and 3 can comprehensively reflect the classification performance while considering the hierarchical relationship among BRCA subtypes. From Supplementary Table 2, we observed that block PLSDA,

block sPLSDA, and MOGONET achieved similar performance on the classification of normal-like vs. non-normal-like subtypes, where MOGONET yielded better ACC and block PLSDA and block sPLSDA yielded better F1 and AUC. However, both block PLSDA and block sPLSDA obtained significantly worse performance than MOGONET when differentiating the different subtypes within the non-normal-like category (Table 4 and Supplementary Table 3). On the other hand, MOGONET consistently outperformed other methods under these three different subtype definitions (Table 4 and Supplementary Tables 2 and 3), which demonstrated that MOGONET could effectively differentiate different BRCA subtypes while considering the intrinsic relationships between different subtypes.

To further demonstrate the generalizability of MOGONET, we also evaluated its performance by training and testing on different patient cohorts from different institutions in the BRCA dataset (Supplementary Table 4). From Supplementary Table 4, we observed that MOGONET achieved similar performance as experiments with randomly partitioned training and test samples, which indicated that MOGONET models could be generalized to different datasets on the same classification task.

**MOGONET outperformed its variations in various classification tasks.** MOGONET combines omics-specific learning via GCNs with cross-omics correlation learning via VCDN for effective multi-omics classification. To examine the necessity of GCN and VCDN for effective multi-omics data classification, we performed extensive ablation studies of our proposed method where three additional variations of MOGONET were compared. (1) NN_NN: fully connected NN with the same number of layers and the same dimensions of hidden layers as the GCN part in MOGONET were used for omics-specific classification. A fully connected NN with the same number of layers as VCDN was used for multi-omics integration. However, instead of constructing the cross-omics discovery tensor, label distribution from each omics data type was directly concatenated to a vector as the input of the multi-omics integration network. (2) NN_VCDN: the omics-specific classification component was the same as NN_NN without utilizing GCNs. The multi-omics integration component utilized VCDN, which was the same as MOGONET. (3) MOGONET_NN: the omics-specific classification component utilized GCN, which was the same as MOGONET. The multi-omics integration part was the same as NN_NN without utilizing VCDN. Note that MOGONET_NN itself is also a novel approach. To the best of our knowledge, there is no existing method that applies GCNs to supervised multi-omics data classification problems.

As shown in Tables 2–4 and Supplementary Table 1, we observed that MOGONET outperformed NN_NN and NN_VCDN in all classification tasks. While MOGONET_NN achieved similar performance as MOGONET in tasks like LGG grade classification, MOGONET still consistently produced better mean metrics than MOGONET_NN in all classification tasks. The similar performance between MOGONET_NN and MOGONET and the better performance of MOGONET_NN than NN_VCDN indicates that our use of GCNs for multi-omics classification tasks makes significant contributions to the performance boost of MOGONET comparing with existing methods. Compared with traditional NN that only learn from omics features, GCNs further exploits the graph structural information within the data. This can be essential to the more comprehensive understanding of the omics data as it captures the connections and correlations among samples. Another possible reason for the similar performance between MOGONET_NN and MOGONET could be related to that the contribution of cross-view correlation in the label space might be limited when the number of distinct categories is small. For example, in the LGG grade classification problem, the number of distinct labels was limited to two. In this case, MOGONET_NN and MOGONET shared the same number of layers for the multi-omics integration component except that the input dimensions were different. In the LGG grade classification problem, the input dimension for the multi-omics integration component was $2 \times 3 = 6$ in MOGONET_NN, while the input dimension for the same component was $2^3 = 8$ in MOGONET. While VCDN can effectively utilize the cross-view correlation in the label space, such advantage could be limited when the number of distinct labels was small. Moreover, in the application of VCDN to human action recognition, its advantage compared with NN became more obvious when dealing with complex datasets with over ten classes[23]. Nevertheless, exploring cross-view correlations was still essential to multi-omics classification as we observed that MOGONET produced better results than MOGONET_NN in all classification tasks under different evaluation metrics. Another interesting observation was that while MOGONET consistently outperformed MOGONET_NN, NN_VCDN failed to consistently outperform NN_NN in all the classification tasks. One possible explanation of this is related to the construction of the cross-omics discovery tensor. Since the input of VCDN was constructed by multiplying the class probabilities predicted by each omics-specific classifier, the prediction noise or error might be amplified if the omics-specific classifiers were not effective. Therefore, GCNs were needed for effective omics-specific learning to fully exploit the advantages of VCDN, and these two components could be trained jointly to achieve superior results for multi-omics classification tasks.

**Performance of MOGONET under different omics data types**. While we used three omics data types in our classification tasks, MOGONET can also be extended to accommodate different numbers of omics data types. To demonstrate the effectiveness of MOGONET with different choices of data modalities, we compared its performance with other methods on the BRCA dataset using only two types of omics data: mRNA expression data and DNA methylation data (Supplementary Table 5). We observed that similar to the case with three different omics data types, MOGONET still consistently outperformed existing methods on the BRCA dataset when trained with mRNA expression and DNA methylation data. This demonstrates that MOGONET could be extendable to different numbers of omics data types.

Moreover, to further demonstrate the necessity of integrating multiple types of omics data to boost the classification performance in biomedical applications, we compared the classification performance of MOGONET with three types of omics data (mRNA + meth + miRNA for combining mRNA expression, DNA methylation, and miRNA expression data), MOGONET with two types of omics data (mRNA + meth for combining mRNA expression and DNA methylation data, mRNA + miRNA for combining mRNA expression and miRNA expression data, and meth + miRNA for combining DNA methylation and miRNA expression data), and the omics-specific GCNs trained with single-omics data type before integration (mRNA for mRNA expression data, meth for DNA methylation data, and miRNA for miRNA expression data). The results are shown in Fig. 2 and Supplementary Fig. 1. From Fig. 2 and Supplementary Fig. 1, we observed that by exploring the cross-omics label correlations through VCDN, the classification performance was consistently improved by integrating classification results from multiple omics data types. Specifically, in all of the classification tasks, MOGONET models trained with three omics data types achieved the best performance comparing with MOGONET models trained with two omics data types. Moreover, all the MOGONET models trained with two omics data types outperformed the single-omics GCN models using the corresponding omics data types. Another interesting observation was that some MOGONET models with two omics data types (e.g., mRNA + miRNA in the ROSMAP dataset, mRNA + meth, and mRNA + miRNA in the BRCA dataset) and omics-specific GCNs (e.g., mRNA GCN in the BRCA dataset) could produce superior results even comparing with some existing multi-omics integration methods trained with three omics data types. This further demonstrates the effectiveness of GCNs in omics data classification problems and the effectiveness of cross-omics learning with VCDN for omics data.

**Performance of MOGONET under different hyper-parameter $k$.** One important hyper-parameter in MOGONET is $k$, which determines the threshold of affinity values adaptively when constructing the weighted sample similarity networks for omics-specific GCNs (Eq. (4)). In our applications, $k$ represents the average number of edges per sample retained in the similarity networks. Similarity networks that faithfully capture the interactions between samples can boost the performance of GCNs by providing additional information on sample correlations. However, if $k$ is too small, the similarity network becomes too sparse, and some important interactions between samples may be missed. In contrast, if $k$ is too large, the similarity network becomes too dense, and noise or artifacts of correlations between samples may be included. Therefore, choosing a proper $k$ value is important for the performance of MOGONET. However, a proper choice of $k$ depends on the topological structure of data, which may vary from dataset to dataset. In our experiments, $k$ was determined through cross-validation on the training data. To further demonstrate the effects of the hyper-parameter $k$ on the performance of MOGONET in both binary and multi-class classification tasks, we trained MOGONET under a wide range of $k$ values using the ROSMAP dataset and the BRCA dataset. Figure 3 shows the performance of MOGONET when $k$ varies from 2 to 10, where the dashed lines represent the results from the best performed existing multi-omics integration methods (GRridge for ROSMAP and XGBoost for BRCA). From Fig. 3, we observed that the hyper-parameter $k$ did influence the classification performance of MOGONET as the performance fluctuated with the change of $k$. However, MOGONET was still robust to the change of $k$ as it consistently outperformed existing methods under different $k$ values. The only exception was when $k = 7$ in the ROSMAP dataset. In this case, while GRridge yielded higher

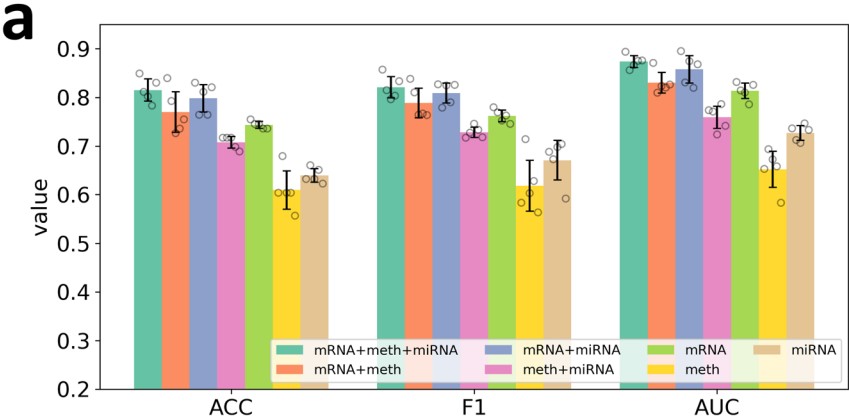

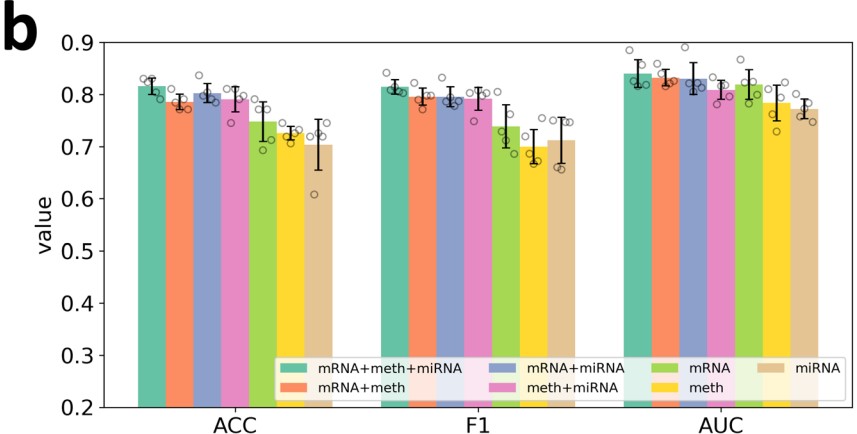

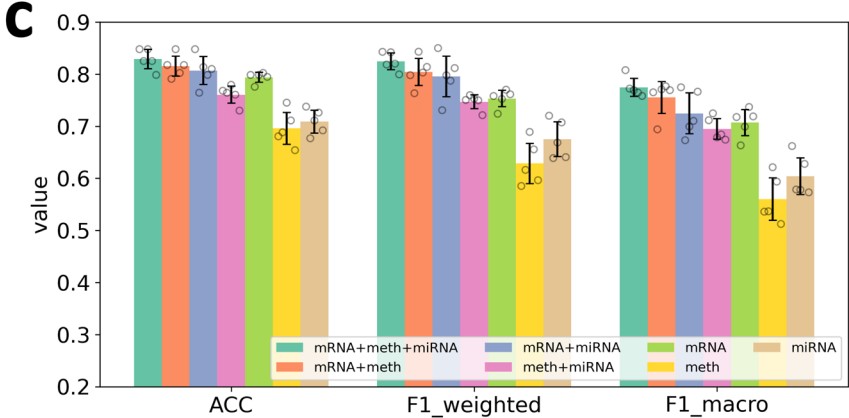

**Fig. 2 Performance comparison of multi-omics data classification via MOGONET and single-omics data classification via GCN (***n* = **5 experiments for each model). a** Results of the ROSMAP dataset. **b** Results of the LGG dataset. **c** Results of the BRCA dataset. Means of evaluation metrics with standard deviations from different experiments are shown in the figure, where the error bar represents plus/minus one standard deviation. mRNA, meth, and miRNA refer to single-omics data classification via GCN with mRNA expression data, DNA methylation data, and miRNA expression data, respectively. mRNA + meth, mRNA + miRNA, and meth + miRNA refer to classification with two types of omics data. mRNA + meth + miRNA refers to classification with three types of omics data. Source data are provided as a Source Data file.

AUC than MOGONET, MOGONET still produced higher ACC and F1 than GRridge.

**Important biomarkers identified by MOGONET.** Following the approach introduced in the "Methods" section, we obtained the rankings of important biomarkers identified by MOGONET. The top 30 important biomarkers for the ROSMAP, BRCA, and LGG dataset were reported in Tables 5 and 6 and Supplementary

Table 8, respectively, with the corresponding rankings in Supplementary Tables 9–11. As mentioned in previous sections, the KIPAN dataset served as a proof-of-concept experiment for multi-class applications, and therefore was excluded from further detailed biomarker identification analysis. Overall, the identified biomarkers by MOGONET were quite diverse within each disease in terms of their function and enriched biological processes. Detailed discussions on the results of the ROSMAP and BRCA datasets are in the following sections, while the discussions on the

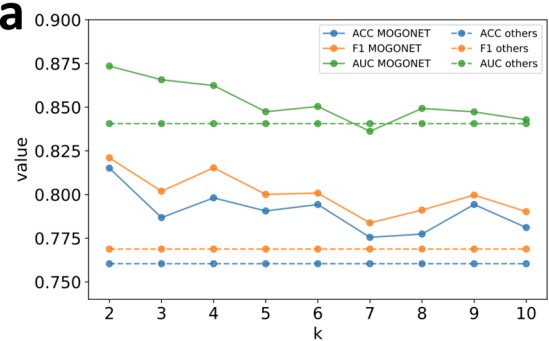
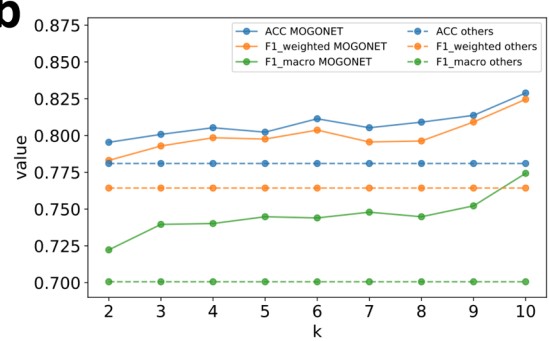

**Fig. 3 Performance of MOGONET under different values of hyper-parameter *k*. a** Results of the ROSMAP dataset. **b** Results of the BRCA dataset. The dashed lines represent the results from the best performed existing multi-omics integration methods (GRridge for ROSMAP and XGBoost for BRCA). MOGONET outperformed the best existing methods under different *k* values. Source data are provided as a Source Data file.

**Table 5 Important omics biomarkers identified by MOGONET in the ROSMAP dataset.**

| Omics data type | Biomarkers |
| --- | --- |
| mRNA expression (8) | NPNT, CDK18, KIF5A, SPACA6, TCEA3, SYTL1, ARRDC2, APLN |
| DNA methylation (5) | TMC4, AGA, HYAL2, CCL3, TTC15 |
| miRNA expression (17) | hsa-miR-423-3p, hsa-miR-33a, hsa-miR-640, hsa-miR-362-3p, hsa-miR-491-5p, hsa-miR-206, hsa-miR-548b-3p, hsa-miR-127-3p, hsa-miR-106a_hsa-miR-17, hsa-miR-424, hsa-miR-577, hsa-miR-873, hsa-miR-651, hsa-miR-199b-5p, hsa-miR-192, hsa-miR-199a-5p, hsv1-miR-H1 |

**Table 6 Important omics biomarkers identified by MOGONET in the BRCA dataset.**

| Omics data type | Biomarkers |
| --- | --- |
| mRNA expression (15) | SOX11, AMY1A, SLC6A15, FABP7, SLC6A14, SLC6A2, FGFBP1, DSG1, UGT8, ANKRD45, PI3, SERPINB5, COL11A2, ARHGEF4, SOX10 |
| DNA methylation (9) | GPR37L1, MIR563, OR1J4, ATP10B, KRTAP3-3, FLJ41941, TMEM207, CDH26, MT1DP |
| miRNA expression (6) | hsa-mir-205, hsa-mir-187, hsa-mir-452, hsa-mir-20b, hsa-mir-224, hsa-mir-204 |

results of the LGG dataset are in the Supplementary Discussion. For comparison purposes, we also used sPLSDA[24] in the mixOmics R package[22] to identify important features in the ROSMAP, BRCA, and LGG datasets (Supplementary Tables 12–14). Unlike block sPLSDA, sPLSDA does not require users to specify the number of identified biomarkers for each omics data type separately, which allows a more direct comparison with MOGONET.

For genes of the top-ranked mRNA expression features and genes inferred from the top-ranked DNA methylation features, we applied the ToppGene Suite[25] for gene set functional enrichment analysis. The enrichment analysis helps us determine if genes identified by MOGONET were biologically meaningful. ToppGene Suite can find biological annotations such as Gene Ontology (GO) terms that are significantly enriched in a set of genes. To account for multiple testings and control the false discovery rate (FDR), the Benjamini–Hochberg procedure was applied, and the adjusted *p* values were reported.

**MOGONET identified biomarkers related to Alzheimer's disease.** For AD patient classification, eight mRNA features, five DNA methylation features, and 17 miRNA features were identified by MOGONET as the top 30 important biomarkers (Table 5). For genes identified by mRNA expression features, several GO terms related to *APLN* and *KIF5A* were significantly enriched, including apelin receptor binding (GO:0031704, $p = 4.90E{-}2$) and central region of growth cone (GO:0090724, $p = 4.82E{-}2$). Moreover, the apelin domain was also significantly enriched by the genes identified from mRNA expression ($p = 1.15E{-}2$). Apelin has been proposed as a promising target for AD[26,27]. Apelin are expressed in various parts of the central nervous system in humans[28] and plays an important role in the pathogenesis of AD[27]. For example, it has been shown that apelin may involve in the regulation of Tau phosphorylation and amyloid-$\beta$ accumulation, thus affects the pathophysiology of AD[26,27,29]. Moreover, as a key isoform of kinesin-1, *KIF5A* is critical in facilitating anterograde mitochondrial transport in neurons[30]. Wang et al.[31] also reported a potential role of *KIF5A* deficiency in AD-relevant axonal mitochondrial traffic abnormalities and suggested therapeutic value in AD treatment through restoring *KIF5A* function. For genes related to the identified DNA methylation features by MOGONET, several GO terms related to the inflammatory process were significantly enriched, including myeloid leukocyte activation (GO:0002274, $p = 3.30E{-}2$), positive regulation of cytokine secretion (GO:0050715, $p = 3.30E{-}2$), and positive regulation of inflammatory response (GO:0050729, $p = 3.30E{-}2$). Several studies have suggested the involvement of inflammation in AD pathogenesis by showing the increased levels of inflammatory cytokines in AD[26,32,33]. It has also been shown that the secretion of cytokines and chemokines could regulate the activity of microglia and astrocytes in AD, which plays a critical role in inflammation and neurodegeneration[34]. On the other hand, for biomarkers identified by sPLSDA, some GO terms related to the solute carrier family were significantly enriched, such as amino acid sodium symporter activity (GO:0005283, $p = 1.56E{-}2$). Moreover, *APLN* and another kinesin-1-related gene *KIF5B* were also identified.

Moreover, highly-ranked genes and miRNAs identified by MOGONET have also been shown to be associated with AD. Cogswell et al.[35] found that the expression level of *hsa-miR-423* was significantly altered in the hippocampus and medial frontal gyrus for early and late-stage AD patients comparing to control samples, where both hippocampus and medial frontal gyrus were regions primarily affected by AD pathology. In addition, Nagaraj et al.[36] reported that *hsa-miR-33a* was deferentially expressed in blood plasma between AD patients and age-matched controls. For identified mRNA expression and DNA methylation

biomarkers, Hohman et al.[37] identified the lower expression of *TMC4* was associated with the amyloid deposition-related decline in executive function. Besides, the overexpression of *CDK18* could regulate the phosphorylation of Tau protein in human brain while hyper-phosphorylated Tau is known to be associated with the pathology of AD[38].

**MOGONET identified biomarkers related to breast cancer**. For BRCA PAM50 subtype classification, 15 mRNA features, nine DNA methylation features, and six miRNA features were identified by MOGONET as the top 30 important biomarkers (Table 6). For genes identified by mRNA expression features, several GO terms related to breast cancer were significantly enriched, including epithelial cell proliferation (GO:0050673, $p = 3.51E-2$) and response to progesterone (GO:0032570, $p = 3.51E-2$). For example, the progesterone receptor is often used as a positive prognostic marker in estrogen receptor-$\alpha$ (ER$\alpha$)+ breast cancers[39]. Mohammed et al.[40] further demonstrated that an activated progesterone receptor could function as a proliferative brake in ER$\alpha$+ breast tumors via modulation of ER$\alpha$ chromatin binding and transcriptional activity. Several GO terms related to the solute carrier family were also significantly enriched, such as neurotransmitter transmembrane transporter activity (GO:0005326, $p = 5.35E-4$) and symporter activity (GO:0015293, $p = 4.35E-3$). Among the identified solute carrier family genes, *SLC6A14* has been shown to be one of the glucose metabolism-related genes that were downregulated by metformin in triple-negative breast cancer (TNBC)[41]. Additionally, prosaposin receptor activity (GO:0036505, $p = 2.69E-2$) was significantly enriched for genes related to the identified DNA methylation features by MOGONET. Wu et al.[42] demonstrated that prosaposin could upregulate estrogen receptor alpha expression through the mitogen-activated protein kinase (MAPK)-signaling pathway and suggested that prosaposin may be involved in breast cancer development and progression. On the other hand, for genes identified from mRNA expression data by sPLSDA, no significantly enriched GO term was found. For DNA methylation features identified by sPLSDA, several biological process terms were significantly enriched, including positive regulation of MAPK cascade (GO:0043410, $p = 4.26E-2$) and regulation of glucose metabolic process (GO:0010906, $p = 4.26E-2$). Studies have shown that hormone-bound steroid receptors activate different complex MAPK-related pathways in breast cancer cells[43,44]. It has also been shown that glucose and factors related to glucose metabolism could contribute to breast cancer development[45].

Moreover, highly-ranked genes and miRNAs identified by MOGONET have also been shown to be associated with breast cancer. For example, Shepherd et al.[46] demonstrated that *SOX11* was critical for regulating the expression of many genes that define the basal-like subtype. They also demonstrated that *SOX11* was related to the invasion and migration of basal-like breast tumors. *FABP7* has also been shown to be related to different breast cancer subtypes. Cordero et al.[47] uncovered the critical role of *FABP7* in metabolic reprogramming of HER2+ breast cancer cells as well as HER2+ breast cancer brain metastasis. Zhang et al.[48] identified a novel subgroup within the basal-like breast tumors with higher expression of *FABP7* that showed significantly better clinical outcomes. For identified miRNA biomarkers, there have been several studies investigating the association between *miRNA-205* and breast cancer. Specifically, *miRNA-205* is generally downregulated and exhibits a tumor-suppressive function in breast cancer[49]. While *miRNA-205* expression is decreased in breast cancer, the relative levels of downregulation vary across different subtypes. For example, *miRNA-205* is upregulated in estrogen/progesterone+ breast cancer compared

with HER2+ breast cancer[50], while TNBCs usually express the least *miRNA-205* among different subtypes[51,52]. Moreover, different studies also showed that metastatic breast cancers had lower expression levels of *miRNA-205* than non-metastatic breast cancers[51,53]. In addition, *miRNA-187* was identified as an independent prognostic factor in breast cancer, where its overexpression was associated with a more aggressive phenotype[54].

**Discussion**

The rapid advancement of omics technologies has enabled personalized medicine using molecular-level data with unprecedented details. Previously, labeled biomedical data have been scarce due to the high expense for collecting and annotating data and the lack of knowledge about subtypes of diseases. Consequently, most existing multi-omics integration methods focus on unsupervised methods without additional phenotypic information and try to extract biological insights from the identified clusters of samples. However, thanks to the rapid development of omics technologies and personalized medicine, as well as large consortium studies such as TCGA and ROSMAP, labeled omics datasets with detailed annotations are becoming available at an unprecedented volume and speed. Therefore, it has become more and more important to take advantage of these labeled omics data to better predict essential phenotypes or traits (e.g., disease diagnosis, grading of tumors, and cancer subtypes) on new samples. To this end, we propose MOGONET, a supervised multi-omics integration method for biomedical classification tasks based on deep multi-view learning, where we consider each omics data type as a view of the samples. We utilized GCNs for omics-specific learning and VCDN for multi-omics integration at the high-level label space. MOGONET also effectively identified meaningful potential biomarkers in each omics data type that showed a strong association with the diseases. To summarize, MOGONET is an innovative deep learning-based multi-omics classification algorithm with both superior performance and good interpretability.

Comparing to fully connected networks, GCNs can utilize both the features and the geometrical structures of the data. While commonly used fully connected networks can only be trained on structured data, GCNs can also generalize NN for arbitrarily structured graphs. This suggests that our GCN-based method is flexible and can be potentially generalized to include more data types to boost the classification performance in the future. We also demonstrated that VCDN can effectively classify multi-omics data by integrating the omics-specific classification produced by GCNs at the label space. As each element in the input of VCDN is constructed by multiplying class probabilities from different classifiers, VCDN might be more sensitive to the noise or error produced in omics-specific learning. Therefore, effective omics-specific classification through GCNs is needed to fully utilize the superiority of VCDN. Through ablation studies, we demonstrated that both GCNs and VCDN were essential to effective multi-omics data classification, while GCNs might play a more essential role in the biomedical classification tasks in this paper.

While we only utilized mRNA expression, DNA methylation, and miRNA expression data for the multi-omics classification tasks in this paper, both the omics-specific GCNs and the multi-omics integration component can be extended to accommodate different or more types of data. Specifically, for a classification task with $c$ classes and $m$ different data types, an individual GCN can be trained for each data type. For multi-omics classification, the label distribution generated by each omic-specific GCN can be integrated by either direct concatenation as in MOGO-NET_NN or constructing a $c^m$-dimensional cross-omics

discovery vector with the similar fashion as in MOGONET. Therefore, MOGONET is a supervised multi-omics classification framework that can be generalized to accommodate many different omics data types.

## Methods

**Overview of MOGONET**. MOGONET is a framework for classification tasks with multi-omics data. The workflow of MOGONET can be summarized into three components: (1) preprocessing. Preprocessing and feature preselection were performed on each omics data type individually to remove noise, artifacts, and redundant features that may deteriorate the performance of the classification tasks. (2) Omics-specific learning via GCNs. For each omics data type, a weighted sample similarity network was constructed from the omics features. Then, a GCN was trained using both the omics features and the corresponding similarity network for omics-specific learning. (3) Multi-omics integration via VCDN. A cross-omics discovery tensor was calculated using the initial class probability predictions from all the omics-specific networks. A VCDN was then trained with the cross-omics discovery tensor to produce the final predictions. VCDN can effectively learn the intra-omics and cross-omics label correlations in the higher-level label space for better classification with multi-omics data. MOGONET is an end-to-end model, where both omics-specific GCNs and VCDN are trained jointly. We describe each component in detail in the following sections.

**Preprocessing**. To remove noise and experimental artifacts in the data and better interpret the results, proper preprocessing for omics data is essential. First, for DNA methylation data, only probes corresponding to the probes in the Illumina Infinium HumanMethylation27 BeadChip were retained for better interpretability of the results. The number of features for each dataset and each omics data type is listed in Table 1. Then, we further filtered out features with no signal (zero mean values) or low variances. Specifically, we applied different variance filtering thresholds for different types of omics data (0.1 for mRNA expression data and 0.001 for DNA methylation data) as different omics data types came with different ranges. For miRNA expression data, we only filtered out features with no variation (variance equals zero) as the available features were limited due to the small number of miRNAs. The same variance thresholds were used across all experiments.

Since omics data could contain redundant features that might have negative effects on the classification performance, we further preselected the omics features through statistical tests. For each classification task, ANOVA $F$-value was calculated sequentially using the training data to evaluate whether a feature was significantly different across different classes, where FDR controlling procedures were applied for multiple-testing compensation. However, selecting too few features might also result in only selecting highly correlated features, which could potentially restrain the models from taking advantage of complementary information from diverse features. To avoid this situation, we determined the number of preselected features for each omics data type with an additional rule, i.e., the first principal component of the data after feature preselection should explain <50% of the variance. We also demonstrated that MOGONET could produce consistent results under a wide range of different numbers of preselected features (Supplementary Fig. 2). The number of preselected features for each dataset is shown in Table 1. Finally, we individually scaled each type of omics data to [0, 1] through linear transformations for training MOGONET.

**GCNs for omic-specific learning**. We utilized GCNs for omic-specific learning in MOGONET, where a GCN was trained for each omics data type to perform classification tasks. While existing GCN models mainly focus on semi-supervised learning by propagating the labels from labeled data to unlabeled ones[55–58], the value of these methods in clinical applications could be limited as the learned GCN model cannot be directly applied to predicting new samples whose data might not be available during the training process. Therefore, in this work, we explored the application of GCN in supervised learning. Our goal is to capture the intrinsic structure of the data through the graphs during network training. This is similar to previous manifold learning approaches to preserve the local information of the data by using a graph regularizer. The benefit of graph NN in a supervised setting is that it not only captures the local intra-class information due to the multi-modal phenomena with each class but also seeks more discriminative features by considering the inter-class information.

By viewing each sample as a node in the sample similarity network, the goal of each GCN in MOGONET is to learn a function of features on a graph $\mathcal{G} = (\mathcal{V}, \mathcal{E})$ to perform classification tasks by utilizing both the features of each node and the relationships between nodes characterized by the graph $\mathcal{G}$. Therefore, a GCN model takes the following two inputs. One input is a feature matrix $\mathbf{X} \in \mathbb{R}^{n \times d}$, where $n$ is the number of nodes and $d$ is the number of input features. The other input is a description of the graph structure, which can be represented in the form of an adjacency matrix $\mathbf{A} \in \mathbb{R}^{n \times n}$. A GCN can be built by stacking multiple

convolutional layers. Specifically, each layer is defined as:

$$\begin{aligned} \mathbf{H}^{(l+1)} &= f(\mathbf{H}^{(l)}, \mathbf{A}) \\ &= \sigma(\mathbf{A}\mathbf{H}^{(l)}\mathbf{W}^{(l)}), \end{aligned} \tag{1}$$

where $\mathbf{H}^{(l)}$ is the input of the $l$th layer and $\mathbf{W}^{(l)}$ is the weight matrix of the $l$th layer. $\sigma(\cdot)$ denotes a non-linear activation function. For effective training of GCNs, Kipf and Welling[55] further modified the adjacency matrix $\mathbf{A}$ as:

$$\widetilde{\mathbf{A}} = \hat{\mathbf{D}}^{-\frac{1}{2}}\hat{\mathbf{A}}\hat{\mathbf{D}}^{-\frac{1}{2}} = \hat{\mathbf{D}}^{-\frac{1}{2}}(\mathbf{A}+\mathbf{I})\hat{\mathbf{D}}^{-\frac{1}{2}}, \tag{2}$$

where $\hat{\mathbf{D}}$ is the diagonal node degree matrix of $\hat{\mathbf{A}}$ and $\mathbf{I}$ is the identity matrix.

In MOGONET, the original adjacency matrix $\mathbf{A}$ is constructed by calculating the cosine similarity between pairs of nodes and edges with cosine similarity larger than a threshold $\epsilon$ are retained. Specifically, $A_{ij}$, which is the adjacency between node $i$ and node $j$ in the graph, is calculated as:

$$A_{ij} = \begin{cases} s(\mathbf{x}_i, \mathbf{x}_j), & \text{if } i \neq j \text{ and } s(\mathbf{x}_i, \mathbf{x}_j) \geq \epsilon \\ 0, & \text{otherwise} \end{cases} \tag{3}$$

where $\mathbf{x}_i$ and $\mathbf{x}_j$ are the feature vectors of node $i$ and node $j$, respectively. $s(\mathbf{x}_i, \mathbf{x}_j) = \frac{\mathbf{x}_i \cdot \mathbf{x}_j}{\|\mathbf{x}_i\|_2 \|\mathbf{x}_j\|_2}$ is the cosine similarity between node $i$ and $j$. The threshold $\epsilon$ is determined given a parameter $k$, which represents the average number of edges per node that are retained including self-connections:

$$k = \sum_{i,j} I(s(\mathbf{x}_i, \mathbf{x}_j) \geq \epsilon) / n, \tag{4}$$

where $I(\cdot)$ is the indicator function and $n$ is the number of nodes. The parameter $k$ for generating the adjacency matrix in Eq. (4) is tuned over {2, 5, 10} with the training data, and the same $k$ value is adopted across all experiments on the same dataset. Note that for $k = 1$, $\mathbf{A}$ will contain no edge, while the final adjacency matrix $\widetilde{\mathbf{A}}$ will only include self-connections. In this case, a GCN will degenerate to a normal fully connected network, while MOGONET will degenerate to NN_VCDN.

Although GCNs have been widely utilized in unsupervised[59–62] and semi-supervised[55–58] learning, in this paper, we further extend the use of GCNs to supervised classification tasks. For training data $\mathbf{X}_{tr} \in \mathbb{R}^{n_{tr} \times d}$, the corresponding adjacency matrix $\widetilde{\mathbf{A}}_{tr} \in \mathbb{R}^{n_{tr} \times n_{tr}}$ can be calculated from Eq. (2). Then, a graph convolutional network $GCN(\cdot)$ can be trained with $\mathbf{X}_{tr}$ and $\widetilde{\mathbf{A}}_{tr}$ and the predictions on the training data can be written as:

$$\hat{\mathbf{Y}}_{tr} = GCN(\mathbf{X}_{tr}, \widetilde{\mathbf{A}}_{tr}), \tag{5}$$

where $\hat{\mathbf{Y}}_{tr} \in \mathbb{R}^{n_{tr} \times c}$. The $i$th row in $\hat{\mathbf{Y}}_{tr}$ represents the predicted label probability of the $i$th training sample and $c$ denotes the number of classes in the classification task. Therefore, both the features and the geometrical structure of the training data are utilized in learning the classification task.

For a new test sample $\mathbf{x}_{te} \in \mathbb{R}^d$, we extend the data matrix to $\mathbf{X}_{trte} = \begin{bmatrix} \mathbf{X}_{tr} \\ \mathbf{x}_{te} \end{bmatrix} \in \mathbb{R}^{(n_{tr}+1) \times d}$ and generate the extended adjacency matrix $\widetilde{\mathbf{A}}_{trte} \in \mathbb{R}^{(n_{tr}+1) \times (n_{tr}+1)}$ according to Eq. (2). Specifically, the entries in the last row and the last column of $\widetilde{\mathbf{A}}_{trte}$ are the only entries calculated during testing to reflect the affinity between the test sample $\mathbf{x}_{te}$ and the training samples $\mathbf{X}_{tr}$. Therefore, given $\mathbf{X}_{trte}$, $\widetilde{\mathbf{A}}_{trte}$ and the trained GCN model $GCN(\cdot)$, we have $\hat{\mathbf{Y}}_{trte} = GCN(\mathbf{X}_{trte}, \widetilde{\mathbf{A}}_{trte}) \in \mathbb{R}^{(n_{tr}+1) \times c}$. The predicted label probability distribution for the test sample is 1111*chetak*, which is the last row of $\hat{\mathbf{Y}}_{trte}$. To this end, both the features of the test sample and the correlations between the test sample and the training samples are utilized in predicting the label of the new test sample $\mathbf{x}_{te}$.

In MOGONET, to perform omic-specific classification, we construct a multi-layer GCN for each omics data type. Specifically, for the $i$th omics data type, a omic-specific GCN, $GCN_i(\cdot)$, is trained with training data $\mathbf{X}_{tr}^{(i)} \in \mathbb{R}^{n_{tr} \times d_i}$ and the corresponding adjacency matrix $\widetilde{\mathbf{A}}_{tr}^{(i)} \in \mathbb{R}^{n_{tr} \times n_{tr}}$. The predictions on the training data can be written as:

$$\hat{\mathbf{Y}}_{tr}^{(i)} = GCN_i(\mathbf{X}_{tr}^{(i)}, \widetilde{\mathbf{A}}_{tr}^{(i)}), \tag{6}$$

where $\hat{\mathbf{Y}}_{tr}^{(i)} \in \mathbb{R}^{n_{tr} \times c}$. We use $\hat{\mathbf{y}}_j^{(i)} \in \mathbb{R}^c$ to denote the $j$th row in $\hat{\mathbf{Y}}_{tr}^{(i)}$, which is the predicted label distribution of the $j$th training sample from the $i$th omics data type. Therefore, the loss function for $GCN_i(\cdot)$ can be written as:

$$L_{GCN}^{(i)} = \sum_{j=1}^{n_{tr}} L_{CE}(\hat{\mathbf{y}}_j^{(i)}, \mathbf{y}_j) = \sum_{j=1}^{n_{tr}} -\log\left(\frac{e^{\hat{\mathbf{y}}_j^{(i)} \cdot \mathbf{y}_j}}{\sum_k e^{\hat{y}_{j,k}^{(i)}}}\right), \tag{7}$$

where $L_{CE}(\cdot)$ represents the cross-entropy loss function. $\mathbf{y}_j \in \mathbb{R}^c$ is the one-hot encoded label of the $j$th training sample and $\hat{y}_{j,k}^{(i)}$ is the $k$th element in the vector $\hat{\mathbf{y}}_j^{(i)}$. Moreover, to account for the label imbalance in the training data, we further apply different weights on the losses of different classes in Eq. (7), where the weight of a class is set to the inverse of its frequency in the training data.

**VCDN for multi-omics integration**. Existing methods utilizing multi-view data for biomedical classification tasks either directly concatenate features from different views or learn to fuse data by learning the weight of each view or fusing features from different views in a low-level feature space[4,63–65]. However, it is always challenging to align various views properly without causing negative influence. On the other hand, VCDN[23] can exploit the higher-level cross-omics correlations in the label space, as different types of omics data can provide unique class-level distinctiveness. VCDN is designed to learn the higher-level intra-view and cross-view correlations in the label space and has shown significant improvements in human action recognition tasks. In MOGONET, we utilize VCDN to integrate different omics data types for classification. Moreover, while the original form of VCDN was designed for samples with two views[23], we further generalize it to accommodate an arbitrary number of data types and demonstrated with three types of omics data: mRNA expression, DNA methylation, and miRNA expression.

Since mRNA expression data, DNA methylation data, and miRNA expression data are used in our experiments, for simplicity, we first demonstrate how to extend VCDN to accommodate three views. For the predicted label distribution of the $j$th sample from different omics data types $\hat{\mathbf{y}}_j^{(i)}$, $i = 1, 2, 3$, we construct a cross-omics discovery tensor $\mathbf{C}_j \in \mathbb{R}^{c \times c \times c}$, where each entry of $\mathbf{C}_j$ is calculated as:

$$C_{j,a_1a_2a_3} = \hat{y}_{j,a_1}^{(1)} \hat{y}_{j,a_2}^{(2)} \hat{y}_{j,a_3}^{(3)}, \tag{8}$$

where $\hat{y}_{j,a}^{(i)}$ denotes the $a$th entry of $\hat{\mathbf{y}}_j^{(i)}$.

Then, the obtained tensor $\mathbf{C}_j$ is reshaped to a $c^3$ dimensional vector $\mathbf{c}_j$ and is forwarded to $VCDN(\cdot)$ for the final prediction. $VCDN(\cdot)$ is designed as a fully connected network with the output dimension of $c$. The loss function of $VCDN(\cdot)$ can be written as:

$$L_{VCDN} = \sum_{j=1}^{n_{tr}} L_{CE}(VCDN(\mathbf{c}_j), \mathbf{y}_j). \tag{9}$$

To this end, $VCDN(\cdot)$ could reveal the latent cross-view label correlations and help to improve the learning performance. By utilizing $VCDN(\cdot)$ to integrate initial predictions from different types of omics data, the final prediction made by MOGONET is based on both the omics-specific predictions and the learned cross-omics label correlation knowledge.

Extension of MOGONET to a different number of views can be performed in a similar fashion. For data with $m$ omics data types, each element in $\mathbf{C}_j$ can be calculated as:

$$C_{j,a_1...a_m} = \prod_{i=1}^{m} \hat{y}_{j,a_i}^{(i)}, \quad a_i = 1, 2, ..., m. \tag{10}$$

Then, the obtained tensor $\mathbf{C}_j$ is reshaped to a $c^m$ dimensional vector, and $VCDN(\cdot)$ can be trained in the same way as Eq. (9).

For MOGONET_NN in the ablation study, the label distribution from each omics data type is directly concatenated to a longer vector as the input of the multi-omics integration network $NN(\cdot)$. The loss function for $NN(\cdot)$ can be written as:

$$L_{NN} = \sum_{j=1}^{n_{tr}} L_{CE}(NN(\mathbf{c}_j'), \mathbf{y}_j) = \sum_{j=1}^{n_{tr}} L_{CE}(NN([\hat{\mathbf{y}}_j^{(1)}, \hat{\mathbf{y}}_j^{(2)}, \hat{\mathbf{y}}_j^{(3)}]), \mathbf{y}_j) \tag{11}$$

where $\mathbf{c}_j' \in \mathbb{R}^{3c}$ is the concatenated vector of the output of the omic-specific GCNs. In our experiments, $NN(\cdot)$ shares the same number of layers as $VCDN(\cdot)$, while the dimensionality of the input data is different.

In summary, in our experiments where three omics data types are used, the total loss function of MOGONET can be written as:

$$L = \sum_{i=1}^{3} L_{GCN}^{(i)} + \gamma L_{VCDN}, \tag{12}$$

where $\gamma$ is a trade-off parameter between the omics-specific classification loss and the final classification loss from $VCDN(\cdot)$. We set $\gamma = 1$ in all our experiments. MOGONET is an end-to-end model, and all networks are trained jointly. For training MOGONET, we first pretrain each of the omics-specific GCN individually to get a good initialization of the GCN. Then, during one epoch of the training process, we first fix $VCDN(\cdot)$ and update $GCN_i(\cdot)$, $i = 1, 2, 3$ for each omics data type to minimize the loss function $L$. Then we fix the omics-specific GCNs and update $VCDN(\cdot)$ to minimize $L$. Omics-specific GCNs and VCDN are updated alternatively until convergence.

**Identifying important biomarkers with MOGONET**. Identifying biomarkers is essential for interpreting the results and understanding the underlying biology in biomedical applications. Over the years, there have been extensive studies on determining the importance of features for NN. Since the input of MOGONET is scaled to [0, 1] during preprocessing, we can remove the signal from a feature by setting it to zero. Therefore, the importance of a feature to the classification task can be measured by the performance decrease after the feature is removed. Such *ablation* approach has been widely adopted for feature importance ranking and feature selection in NN[2,66,67]. Using this approach, we analyzed the contribution of each feature in different types of omics data by assigning the feature to zero and calculating the classification performance decrease on the test set comparing to using all the features. Features with the largest performance drop were considered

to be the most important ones. We used the F1 score to measure the performance drop for binary classification tasks and F1_macro for multi-class classification tasks. To account for the randomness in the training process, we performed five repeated experiments in a dataset and summarized the results by summing up the performance decrease in the repeated experiments. Since each omics data type carried the same importance in MOGONET during the construction of the cross-omics discovery tensor, while different numbers of features were preselected for training in different omics data types, we further scaled the feature importance of each omics data type by the number of preselected features. Finally, we ranked the importance of each feature.

**Reporting summary**. Further information on research design is available in the Nature Research Reporting Summary linked to this article.

## Data availability

The ROSMAP dataset was obtained from AMP-AD Knowledge Portal (https://adknowledgeportal.synapse.org/). Omics data of LGG, KIPAN, and BRCA, as well as the grade information of LGG patients, were obtained from The Cancer Genome Atlas Program (TCGA) through Broad GDAC Firehose (https://gdac.broadinstitute.org/). PAM50 breast cancer subtypes of TCGA BRCA patients were obtained through the TCGAbiolinks R package (v2.12.6, http://bioconductor.org/packages/release/bioc/html/TCGAbiolinks.html). Source data are provided with this paper.

## Code availability

The source code of this work can be downloaded from GitHub[68] (https://github.com/txWang/MOGONET).

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

## Acknowledgements

This work was supported by National Institutes of Health grants R01EB025018 (K.H., T.W.) and U54AG065181 (K.H., J.Z., W.S.); and Indiana University Precision Health Initiative (J.Z.).

## Author contributions

T.W., Z.D., and K.H. conceived and designed the study. T.W. and W.S. performed the computational analysis with assistance from Z.H. and J.Z. T.W., Z.D., and K.H. wrote the manuscript. W.S., Z.H., H.T., and J.Z. edited the manuscript. All the authors reviewed and approved the final manuscript.

## Competing interests

The authors declare no competing interests.
