## [Peer Review File · Nature Communications]

Reviewers' Comments:

Reviewer #1:

Remarks to the Author:

The paper is interesting but I find some critical points that are an obstacle to the publication of the paper.

1. The model presented by the authors is a generalization of a previous model from 2 omics to 3 omics.
2. Although there are few applications of supervised methods for classification in multi omics data, this is not true in many other applications
3. The biological findings of the paper are just a confirmation of previous biomarkers and the number of biomarkers is too high
4. There is a missing of comparison with other inferential methodologies in the lists of biomarkers.
5. There is not a ranking of the biomarkers and a justification of them (from an inferential point of view).
6. There are some lacks and not well presented results (for example, missing of discussion about methylation data, or Go annotations in Alzheimer disease)

Reviewer #2:

Remarks to the Author:

Wang et al.'s article addresses a relevant, unmet need for the integrated analysis of multi-omics datasets in a supervised learning setting.

The proposed method, MORONET, is sufficiently original and sound to merit its review in this journal.

Despite these strengths, the following weaknesses are identified:

1. The article focuses on application cases consisting of three datasets only. Although the method is in principle extendable to other analysis scenarios, it is important to examine at least one application case involving other numbers of input datasets, e.g., 2 input datasets.
2. For the application cases already included in the article, comparisons of performance between 3-dataset vs. 2-dataset vs. single-source models are needed.
3. To strengthen the claims of generalizability and potential biomedical relevance of the proposed method, it would be important to include (at least for one of the application cases presented) a further evaluation using a second test dataset that is fully independent from the one used for development and initial evaluation.
4. All the analyses are based on models with 200 inputs for each omic dataset. What is the effect of using different numbers of inputs?
5. Performance evaluation for the multi-class model is reported using ACC values only. AUC estimates for multi-class performance are also required for an objective assessment of these models.
6. The claim that, in comparison with other methods, MORONET "was still able to produce more consistent results" was not convincingly supported. Further systematic, statistical support beyond a look at SDs values would be needed to adequately address this comparative performance aspect.
7. The assessment of the relevance of the candidate biomarkers was not sufficiently convincing. Although the article offers several examples of the potential biological relevance of individual genes based on published literature, statistical support is needed for allowing a more objective and

systematic evaluation. For instance, a gene set-level statistical assessment of biological annotations is necessary in all application cases (as done in the case of the breast cancer study).

8. The article did not present evidence of the robustness of the proposed biomarker identification approach. An analysis of the feature selection stability for multiple training-testing data samplings is important to understand the strengths and limitations of the proposed methodology.

9. The shared code does not include the implementation of the biomarker selection strategy. This code is necessary for supporting the reproducibility of results during and after review and for enabling other future research.

10. The description of the VCDN concept and its advantages in the Discussion was not sufficiently clear. Additional information about VCDN is also needed in Methods.

11. It is unclear why the performance of MORONET under different values of hyper-parameter k was presented in the Discussions. The Results section would be a more appropriate section for this analysis. Further details, that could be included in a supplementary section, would also be required anyway for strengthening these observations and making recommendations for users on the selection of this critical parameter.

12. The article offers code and example data to reproduce the classification results of the breast cancer application. To allow users, including the reviewers, to reproduce all reported results and facilitate future research, there is a need for a general-purpose or more flexible working implementation of MORONET. Such an implementation should allow users to implement different models, e.g., different numbers of input datasets. In its current form, the shared code allows a limited reproduction of one aspect of the research (classification only) for only one application case.

13. All figure captions need more detailed and clear descriptions. For example, what is the meaning of the error bars?

14. Many parts of the article, in particular Results and Discussion, are not clearly written and contain grammatical errors and lack clarity. Overall, the article deserves a careful revision to improve clarity and quality of writing.

Reviewer #3:

Remarks to the Author:

In this paper, the authors proposed Multi-Omics gRaph cOnvolutional NETworks (MORONET), a multi-omics analysis framework for performing classification in biomedical applications. MORONET first trained graph convolutional networks (GCN) using patient similarity networks based on multiple types of omics data. A cross-omics discovery tensor was calculated from the omics specific predictions and forwarded to view correlation discovery network (VCDN) for multi-omics integrative classification. The framework benefits from taking advantage of the correlation structure among patients (GCNs) and across different omics data types (VCDN). It identified different biomarkers and, in most cases, outperformed various supervised learning methods when applied to four different diseases (Alzheimer's disease, low grade glioma, kidney cancer and breast invasive carcinoma) and three types of omics data (mRNA expression, DNA methylation, and miRNA expression). The addressed question is relevant and interesting to the bioinformatics community, and the text is easy to read. However, the reviewer has the following concerns:

1. The authors seemed to have adopted a flat classification approach when dealing with breast invasive carcinoma (BRCA) subtypes, whereas the considered classes were actually hierarchical. One of the categories was normal and the other four were BRCA subtypes. Among the subtypes, two were

luminal subtypes (LumA and LumB) and should be more associated with each other than with, for example, basal-like tumors. When evaluating the performance of different algorithms, flat-classification-based accuracy (ACC) and weighted F1 score (F1) may not be most meaningful to a clinical setting for BRCA.

2. For multi-class classification problems, the authors used ACC and weight averaged F1 to evaluate the performance of different classifiers. Since the data sets have imbalanced classes, it is better to include macro-averaged F1 as well.

3. When identifying biomarkers, the authors "used AUC to measure the performance drop for binary classification tasks and ACC for multi-class classification tasks". ACC is prone to class imbalance for multi-class classification tasks, macro-averaged F1 is probably better.

4. The authors performed gene set enrichment analysis and reported the p-values. The authors mentioned that they performed adjustments for multiple testing but did not elaborate what method they used. Additionally, it was unclear whether they reported adjusted or unadjusted p-values. The authors should describe how they corrected for multiple testing and report the adjusted p-values.

5. The data pre-processing part only kept the omics features most significantly different across different classes in the training set. Since omics features could be highly correlated, this approach risked selecting a single cluster of omics features, whereas the algorithm will potentially benefit from less inter-correlated features leading different clusters. This pre-processing step may also introduce potential issues in the biomarker discovery part.

6. From the results, NN_NN often outperformed NN_VCDN whereas for GCN_NN and GCN_VCDN (MORONET) it was the opposite. It's unclear if the improvement brought by VCDN was actually dependent on the omics-specific part. The authors did mention that "... Another interesting observation was that while MORONET consistently outperformed NN_NN, both NN_VCDN and GCN_NN failed to consistently outperformed NN_NN in all the tasks, which suggested that GCN and VCDN need to be combined and trained jointly in order to achieve superior results for multi-omics classification tasks." This statement is superficial: GCN_NN only performed worse than NN_NN in the KIPAN data set -- the easiest data set for classification, GCN_NN are often close to MORONET. The authors should go into more detailed comparisons in the discussion part to clarify the advantage of VCDN.

Minor issues/comments:

1. The abbreviations of subtypes of BRCA were first introduced in Table 1 without mentioning their full form. This can be confusing to a non-specialist audience.

2. In general, acronyms should be explained in captions of tables to be self-contained.

3. Strictly speaking, ACC, F1 and area under the curve (AUC) should be decimals. All the reported numbers in this paper were multiplied by 100.

4. The authors compared MORONET with random forests. It may be helpful to include other tree-based ensemble methods like extreme gradient boosting in the comparison as well.

5. Page 6: "For KIPAN, while results of MORONET were slightly better than only using DNA methylation data." This sentence seemed to be incomplete and should be combined with the following sentence.

6. Figure 2 did not state clearly the meaning of error bars in its legend. Is it plus/minus one standard deviation or a 95% confidence interval? The former was used in the paper's tables but in visualization the latter is more widely used.

7. In Figure 3, the authors claimed that "MORONET consistently outperformed existing methods under a wide range of k values". This was not entirely true for ROSMAP data when AUC was the evaluation criterion. The statement needs to be toned down.

Response to Reviewer 1

Comment 1: *The model presented by the authors is a generalization of a previous model from 2 omics to 3 omics.*

Response: Thank you for this comment. While the original application of VCDN algorithm [1] was designed for human action recognition tasks with only two views, MORONET was designed for multi-omics classification tasks and can be extended to handle arbitrary numbers of omics data types. To best accommodate the learning on omics data, we innovatively took advantage of the GCNs for omics-specific learning, which can explicitly exploit both the omics features and the correlations between samples and enables graph-based learning. MORONET is the first method that utilizes GCNs in performing multi-view learning with VCDN. Moreover, while the existing applications of GCNs were mainly designed for unsupervised or semi-supervised tasks, we further extended the use of GCNs to supervised tasks in biomedical problems. To the best of our knowledge, MORONET is also the first supervised multi-omics integrative method for effective class prediction on new samples that utilizes GCNs for omics data learning. In summary, our proposed method is not only a simple generalization of the existing models.

Comment 2: *Although there are few applications of supervised methods for classification in multi omics data, this is not true in many other applications.*

Response: We thank the reviewer for this comment. With the rapid development in omics technologies and personalized medicine, curated datasets with detailed annotations to characterize the phenotypes or traits of the samples are becoming available. Therefore, as the reviewer pointed out, while there have not been many studies on supervised method development, there is an increasing interest in supervised multi-omics integration methods that can perform predictions on new samples and identify disease-related biomarkers. Moreover, while existing applications of GCNs mainly focused on unsupervised or semi-supervised learning tasks, we innovatively extended the use of GCNs to supervised learning tasks. Compared to existing methods, GCNs can further take advantage of the intrinsic structure within the data to boost the classification performance. Therefore, we consider that MORONET can not only addresses the interest in the development of supervised multi-omics integrative analysis methods, but also carries novelty and insights in terms of the algorithm design.

Comment 3: *The biological findings of the paper are just a confirmation of previous biomarkers and the number of biomarkers is too high.*

Response: We thank the reviewer for this insightful comment. Besides relying on existing experimental studies to discuss the identified biomarkers, we also performed gene set functional enrichment analysis to determine if genes identified by MORONET were biologically relevant. In this revision we included a ranking scheme for the identified biomarkers so that the users can choose the number of the most relevant biomarkers according to their applications. In our experiments, the top 30 biomarkers for each dataset were reported in Tables 5-7 with their corresponding rankings in Tables S8-S10. Overall, the identified biomarkers by MORONET were quite diverse within each disease in terms of their function and enrichment, and we included detailed discussions of each dataset in the section “Important biomarkers identified by MORONET”.

Comment 4: *There is a missing of comparison with other inferential methodologies in the lists of biomarkers.*

Response: Based on the reviewer’s suggestion, we tested sPLSDA [2] in the mixOmics R package [3] to identify important features in the ROSMAP, LGG, and BRCA datasets (Tables S11-S13) for comparison. Specifically, while the biomarkers identified by sPLSDA and MORONET were quite different, MORONET yielded richer and more diverse biomarkers within each disease in terms of their function, enrichment, and omics data type. We included the detailed discussions of the identified biomarkers by each method in the section “Important biomarkers identified by MORONET”.

Comment 5: *There is not a ranking of the biomarkers and a justification of them (from an inferential point of view).*

Response: We thank the reviewer for this great comment. We have included the rankings of biomarkers based on previous approaches that determine the importance of a feature to the classification task by measuring the performance decrease after the feature is removed. The rankings of the top 30 important biomarkers identified by MORONET in the ROSMAP, LGG, and BRCA datasets are shown in Tables S8-S10.

Comment 6: *There are some lacks and not well presented results (for example, missing of discussion about methylation data, or Go annotations in Alzheimer disease).*

Response: Thank you for this suggestion. We have included detailed discussions about the highly-ranked mRNA, DNA methylation, and miRNA biomarkers of each dataset in the section “Important biomarkers identified by MORONET”. We also included the discussions about the gene set functional enrichment analysis of the identified biomarkers in the ROSMAP dataset in the section “MORONET identified biomarkers related to Alzheimer's disease”.

Response to Reviewer 2

Comment 1: *The article focuses on application cases consisting of three datasets only. Although the method is in principle extendable to other analysis scenarios, it is important to examine at least one application case involving other numbers of input datasets, e.g., 2 input datasets.*

Response: We thank the reviewer for this insightful comment. We have added a comparison of MORONET with existing methods on the BRCA dataset with two types of omics data (mRNA expression data and DNA methylation data, Table S5). We observed that similar to the case with three different omics data types, MORONET still consistently outperformed existing methods on the BRCA dataset when trained with only mRNA expression and DNA methylation data.

Comment 2: *For the application cases already included in the article, comparisons of performance between 3-dataset vs. 2-dataset vs. single-source models are needed.*

Response: We thank the reviewer for this great suggestion. We have included comparisons of MORONET with one, two, or three omics data types (Figure 2 and S1). Specifically, MORONET with only one omics data type degenerates to an omics-specific GCN. We observed that MORONET models trained with three omics data types achieved the best performance compared to MORONET models with two omics data types. Moreover, each of the MORONET models trained with two omics data types outperformed the two single-omics GCN models trained with the corresponding omics data types.

Comment 3: *To strengthen the claims of generalizability and potential biomedical relevance of the proposed method, it would be important to include (at least for one of the application cases presented) a further evaluation using a second test dataset that is fully independent from the one used for development and initial evaluation.*

Response: Thank you for this insightful comment. While currently large-scale multi-omics datasets for diseases are still rare, to further demonstrate the generalizability of MORONET, we evaluated its performance by training and testing on samples from separate institutions in the BRCA dataset (Table S4). The BRCA dataset contains samples obtained from multiple sites. We choose 577 samples from 28 sites for training the model, and then directly test the model on 298 sample from another 12 sites without any additional training. We observed that MORONET achieved similar performance as experiments with randomly partitioned training and testing samples (Table 4), which indicated that MORONET models could potentially be generalized to data from different sources on the same classification task.

Comment 4: *All the analyses are based on models with 200 inputs for each omic dataset. What is the effect of using different numbers of inputs?*

Response: Thanks for this important comment. Based on the Reviewer 3's comment, we have revised our approach for pre-processing the data, and the number of features selected after pre-processing for each omics type is listed in Table 1. Moreover, we compared the performance of MORONET with the best existing method (XGBoost) in the BRCA dataset with different numbers of inputs (Figure S2). We observed that MORONET could produce consistent results under a wide range of different numbers of pre-selected features. MORONET also consistently outperformed the best existing method under a wide range of different numbers of pre-selected features in the BRCA dataset.

Comment 5: *Performance evaluation for the multi-class model is reported using ACC values only. AUC estimates for multi-class performance are also required for an objective assessment of these models.*

Response: Thank you for this comment. For multi-class classification tasks, besides accuracy (ACC), we also reported the average F1 score weighted by support (F1_weighted) and the macro-averaged F1 score (F1_macro) for a comprehensive assessment of the classification performance. Moreover, we also evaluated the performance of multi-class classification tasks using the average AUC score weighted by support (AUC_weighted) and the macro-averaged AUC score (AUC_macro), with the results for the BRCA and KIPAN dataset shown in Tables S6-S7.

Comment 6: *The claim that, in comparison with other methods, MORONET “was still able to produce more consistent results” was not convincingly supported. Further systematic, statistical support beyond a look at SDs values would be needed to adequately address this comparative performance aspect.*

Response: We thank the reviewer for this comment. We have modified our framework based on the reviewers' comments and re-ran the experiments. Under our previous framework, we observed that MORONET yielded similar performance when using three different omics data and using only DNA methylation data in the KIPAN dataset, while MORONET with three different omics data yielded smaller standard deviation. However, under our current framework, MORONET with three different omics data produce higher average performance than MORONET with only DNA methylation data in the KIPAN dataset when evaluated using accuracy, F1 score weighted by support, and macro-averaged F1 score. Therefore, in the updated results, we focus our discussion on the improved performance and removed this statement from the manuscript.

Comment 7: *The assessment of the relevance of the candidate biomarkers was not sufficiently convincing. Although the article offers several examples of the potential biological relevance of individual genes based on published literature, statistical support is needed for allowing a more objective and systematic evaluation. For instance, a gene set-level statistical assessment of biological annotations is necessary in all application cases (as done in the case of the breast cancer study).*

Response: Thank you for this important comment. We have included the discussions about the gene set functional enrichment analysis of the identified biomarkers in each of the dataset in the section “Important biomarkers identified by MORONET”.

Comment 8: *The article did not present evidence of the robustness of the proposed biomarker identification approach. An analysis of the feature selection stability for multiple training-testing data samplings is important to understand the strengths and limitations of the proposed methodology.*

Response: We thank the reviewer for this insightful comment. We evaluated the robustness of the biomarker identification approach by comparing the identified top 50 important biomarkers in different experiments. Specifically, for experiments with different training and testing partitions, the pre-selected features in each experiment could be different. To compare the two sets of identified biomarkers between two experiments, we use n_1 and n_2 to denote the number of features in the dataset in the first and second experiment, respectively. We use m_1 and m_2 to denote the number of identified biomarkers in the first and second experiment, respectively. We use o to denote the number of overlapping features between the datasets in the first and second experiment, and use o_b to denote the number of overlapping identified biomarkers between the first and second experiment. Therefore, the probability that the number of overlapping identified

biomarkers between these two experiments is larger or equal to can be calculated in an exact form as:

$$= \frac{\sum_{i=1}^{\min(1,2)} \binom{m_i}{i}}{\binom{m_1+m_2}{m_1}}$$

This could be also considered as the p value that determines whether the two sets of identified biomarkers are significantly similar or share a significantly large number of overlapping biomarkers.

We performed four additional experiments with four different training and testing partitions on the ROSMAP and the BRCA dataset, respectively. We compared the identified biomarkers in these additional experiments with the top 50 biomarkers in the experiments reported in the manuscript by calculating the p values. For the ROSMAP dataset, the p values were 1.43E-14, 3.38E-29, 5.03E-14, and 8.60E-20. For the BRCA dataset, the p values were 2.50E-33, 7.58E-40, 1.68E-27, and 1.72E-37. This demonstrates that MORONET can produce significantly similar biomarkers under different experiments with different training and testing partitions.

Comment 9: *The shared code does not include the implementation of the biomarker selection strategy. This code is necessary for supporting the reproducibility of results during and after review and for enabling other future research.*

Response: We thank the reviewer for this great suggestion. We have uploaded the code for identifying the important biomarkers (<https://github.com/txWang/MORONET>).

Comment 10: *The description of the VCDN concept and its advantages in the Discussion was not sufficiently clear. Additional information about VCDN is also needed in Methods.*

Response: Thank you for this insightful comment. We have included more discussions about VCDN in the Results section (Lines 201-207) and the Discussion section (Lines 391-398). Specifically, through ablation studies, we demonstrated that both GCNs and VCDN were effective for the multi-omics data classification task, while GCNs might play a more essential role in the biomedical classification tasks we evaluated in this paper. However, as each element in the input of VCDN is constructed by multiplying class probabilities from different classifiers, VCDN might be more sensitive to the noise or error produced in omics-specific learning. Therefore, effective omics-specific classification through GCNs is needed to fully utilize the superiority of VCDN.

We also included more details about VCDN, as well as its application to different numbers of omics data types in the Methods section (Lines 514-516).

Comment 11: *It is unclear why the performance of MORONET under different values of hyper-parameter k was presented in the Discussions. The Results section would be a more appropriate section for this analysis. Further details, that could be included in a supplementary section, would also be required anyway for strengthening these observations and making recommendations for users on the selection of this critical parameter.*

Response: We have moved the results about the hyper-parameter k to the section “Performance of MORONET under different hyper-parameter k ” within the results section.

Comment 12: *The article offers code and example data to reproduce the classification results of the breast cancer application. To allow users, including the reviewers, to reproduce all reported results and facilitate future research, there is a need for a general-purpose or more flexible working implementation of MORONET. Such an implementation should allow users to implement different models, e.g., different numbers of input datasets. In its current form, the shared code allows a limited reproduction of one aspect of the research (classification only) for only one application case.*

Response: We thank the reviewer for this great comment. We have revised our code to handle arbitrary numbers of omics data types, which can be determined by the users. We also included data and code for multiple datasets, as well as the code for identifying the important biomarkers (<https://github.com/txWang/MORONET>).

Comment 13: *All figure captions need more detailed and clear descriptions. For example, what is the meaning of the error bars?*

Response: We have included detailed and clear descriptions of each figure in this revision. The error bar in Figure 2 represents plus/minus one standard deviation of the results from different experiments.

Comment 14: *Many parts of the article, in particular Results and Discussion, are not clearly written and contain grammatical errors and lack clarity. Overall, the article deserves a careful revision to improve clarity and quality of writing.*

Response: We have carefully edited and revised the manuscript.

Response to Reviewer 3

Comment 1: *The authors seemed to have adopted a flat classification approach when dealing with breast invasive carcinoma (BRCA) subtypes, whereas the considered classes were actually hierarchical. One of the categories was normal and the other four were BRCA subtypes. Among the subtypes, two were luminal subtypes (LumA and LumB) and should be more associated with each other than with, for example, basal-like tumors. When evaluating the performance of different algorithms, flat-classification-based accuracy (ACC) and weighted F1 score (F1) may not be most meaningful to a clinical setting for BRCA.*

Response: We thank the reviewer for this insightful comment. Since the different PAM50 subtypes in BRCA carry different meanings with some categories being more similar than others in clinical settings, we further evaluated the performance of different methods with the following two additional definitions of labels while trained with the labels of five subtypes. One was the binary classification of normal-like vs. non-normal-like subtypes, where the non-normal-like categories included the rest four different subtypes (Table S2). The other one included four categories, where the Luminal A and Luminal B subtypes were merged into one category as they should be more associated with each other than the rest of the subtypes as suggested by the reviewer (Table S3). We also included the detailed discussion of the results in Lines 154-165.

Comment 2: *For multi-class classification problems, the authors used ACC and weight averaged F1 to evaluate the performance of different classifiers. Since the data sets have imbalanced classes, it is better to include macro-averaged F1 as well.*

Response: Thank you for this great suggestion. For a comprehensive assessment of the classification performance, we used accuracy (ACC), F1 score (F1), and area under the receiver operating characteristic curve (AUC) for binary classification tasks, and used accuracy (ACC), average F1 score weighted by support (F1_weighted), and macro-averaged F1 score (F1_macro) for multi-class classification tasks.

Comment 3: *When identifying biomarkers, the authors “used AUC to measure the performance drop for binary classification tasks and ACC for multi-class classification tasks”. ACC is prone to class imbalance for multi-class classification tasks, macro-averaged F1 is probably better.*

Response: We thank the reviewer for this great suggestion. In our revised manuscript, we used F1 score to measure the performance for binary classification tasks and used macro-averaged F1 score (F1_macro) for multi-class classification tasks.

Comment 4: *The authors performed gene set enrichment analysis and reported the p-values. The authors mentioned that they performed adjustments for multiple testing but did not elaborate what method they used. Additionally, it was unclear whether they reported adjusted or unadjusted*

p-values. The authors should describe how they corrected for multiple testing and report the adjusted p-values.

Response: To account for multiple tests and control the false discovery rate, Benjamini–Hochberg procedure was applied and the adjusted p values were reported. We clarified this in the revised manuscript.

Comment 5: *The data pre-processing part only kept the omics features most significantly different across different classes in the training set. Since omics features could be highly correlated, this approach risked selecting a single cluster of omics features, whereas the algorithm will potentially benefit from less inter-correlated features leading different clusters. This pre-processing step may also introduce potential issues in the biomarker discovery part.*

Response: We thank the reviewer for this insightful comment. Based on this comment, we have revised our approach for pre-processing the data. To avoid only selecting highly correlated features, we determined the number of pre-selected features for each omics data type with an additional rule, *i.e.*, the first principal component of the data after pre-processing should explain less than 50% of the variance. The number of features selected after pre-processing for each dataset and each omics type is listed in Table 1.

Moreover, we compared the performance of MORONET with the best existing method (XGBoost) in the BRCA dataset with different numbers of inputs (Figure S2). We observed that MORONET could produce consistent results under a wide range of different numbers of pre-selected features. MORONET also consistently outperformed the best existing method under a wide range of different numbers of pre-selected features in the BRCA dataset.

Comment 6: *From the results, NN_NN often outperformed NN_VCDN whereas for GCN_NN and GCN_VCDN (MORONET) it was the opposite. It's unclear if the improvement brought by VCDN was actually dependent on the omics-specific part. The authors did mention that "... Another interesting observation was that while MORONET consistently outperformed NN_NN, both NN_VCDN and GCN_NN failed to consistently outperformed NN_NN in all the tasks, which suggested that GCN and VCDN need to be combined and trained jointly in order to achieve superior results for multi-omics classification tasks." This statement is superficial: GCN_NN only performed worse than NN_NN in the KIPAN data set -- the easiest data set for classification, GCN_NN are often close to MORONET. The authors should go into more detailed comparisons in the discussion part to clarify the advantage of VCDN.*

Response: Thank you for this insightful comment. We have included more discussions about VCDN in the Results section (Lines 201-207) and the Discussion section (Lines 391-398). Specifically, through ablation studies, we demonstrated that both GCNs and VCDN can boost the task for multi-omics data classification, while GCNs might play a more essential role in the biomedical classification tasks we evaluated in this paper. However, as each element in the input

of VCDN is constructed by multiplying class probabilities from different classifiers, VCDN could be more sensitive to the noise or error produced in omics-specific learning. Therefore, effective omics-specific classification through GCNs is needed to fully utilize the superiority of VCDN.

Minor comment 1: *The abbreviations of subtypes of BRCA were first introduced in Table 1 without mentioning their full form. This can be confusing to a non-specialist audience.*

Response: We have included the full form of the name for each subtype in the main text and in the caption of Table 1.

Minor comment 2: *In general, acronyms should be explained in captions of tables to be self-contained.*

Response: We have revised the captions of the tables accordingly as suggested.

Minor comment 3: *Strictly speaking, ACC, F1 and area under the curve (AUC) should be decimals. All the reported numbers in this paper were multiplied by 100.*

Response: We have revised the reported numbers accordingly.

Minor comment 4: *The authors compared MORONET with random forests. It may be helpful to include other tree-based ensemble methods like extreme gradient boosting in the comparison as well.*

Response: We thank the reviewer for this great suggestion. We have included the results of the gradient tree boosting-based classifier implemented in the XGBoost package for comparison.

Minor comment 5: *Page 6: "For KIPAN, while results of MORONET were slightly better than only using DNA methylation data." This sentence seemed to be incomplete and should be combined with the following sentence.*

Response: Thank you for this suggestion. We have revised the manuscript accordingly.

Minor comment 6: *Figure 2 did not state clearly the meaning of error bars in its legend. Is it plus/minus one standard deviation or a 95% confidence interval? The former was used in the paper's tables but in visualization the latter is more widely used.*

Response: The error bar in Figure 2 represents plus/minus one standard deviation of the results from different experiments. We have included detailed descriptions in the caption of this figure in the revised manuscript.

Minor comment 7: *In Figure 3, the authors claimed that “MORONET consistently outperformed existing methods under a wide range of k values”. This was not entirely true for ROSMAP data when AUC was the evaluation criterion. The statement needs to be toned down.*

Response: Thank you for this comment. We have revised the manuscript accordingly (Lines 243249).

Reference

1. Wang, L., Ding, Z., Tao, Z., Liu, Y. & Fu, Y. Generative multi-view human action recognition. In Proceedings of the IEEE International Conference on Computer Vision, 6212–6221 (2019).
2. Lê Cao, K.-A., Boitard, S. & Besse, P. Sparse pls discriminant analysis: biologically relevant feature selection and graphical displays for multiclass problems. *BMC bioinformatics* **12**, 253 (2011).
3. Rohart, F., Gautier, B., Singh, A. & Lê Cao, K.-A. mixomics: An R package for 'omics feature selection and multiple data integration. *PLoS computational biology* **13**, e1005752 (2017).

Reviewers' Comments:

Reviewer #1:

Remarks to the Author:

The authors answered all my questions in a reasonable way, strongly improving the paper.

Reviewer #2:

Remarks to the Author:

I am satisfied that the authors adequately addressed all my comments and concerns.

Reviewer #3:

Remarks to the Author:

The authors have addressed most of the reviewer's concerns. The remaining ones are as follows.

Following up on Comment 1

It is interesting that judging from Table S2, their method's performance is not that good when classifying normal-like vs non-normal-like. Additionally, I find the new evaluation approach used in Table S3 still not fully addressing the original comment, since they are still treating the four as "flat" categories when the relationship is hierarchical. Instead of "Normal-like vs. Basal-like vs. Her2-enriched vs. Luminal (LumA and LumB)" it is more like "Normal-like vs. BRCA subtypes [Basal-like vs. Her2-enriched vs. Luminal (LumA and LumB)]".

Following up on Comment 5

Their current way of solving the problem is adding a rule for the number of pre-selected features (the first principal component of the data after pre-processing should explain less than 50% of the variance). Personally, I feel that in this way, if there is a large number of inter-correlated features, the highly correlated ones will still be selected -- it is just that a larger number of features will be selected in total. A more natural way is to prune/clump the features before feature-selection.

Response to Reviewer 3

Following up on comment 1: *It is interesting that judging from Table S2, their method's performance is not that good when classifying normal-like vs non-normal-like. Additionally, I find the new evaluation approach used in Table S3 still not fully addressing the original comment, since they are still treating the four as "flat" categories when the relationship is hierarchical. Instead of "Normal-like vs. Basal-like vs. Her2-enriched vs. Luminal (LumA and LumB)" it is more like "Normal-like vs. BRCA subtypes [Basal-like vs. Her2-enriched vs. Luminal (LumA and LumB)]".*

Response: We thank the reviewer for this comment. For BRCA subtype classification, all the compared methods were trained to predict five different subtypes (Normal-like, Basal-like, HER2-enriched, Luminal A, and Luminal B). However, to comprehensively evaluate the classification performance with the consideration of the hierarchical relationship among different BRCA subtypes, for the same method, we performed three different evaluations with three different definitions of labels (Normal-like vs. Basal-like vs. HER2-enriched vs. Luminal A vs. Luminal B for Table 4, Normal-like vs. Non-normal-like for Table S2, and Normal-like vs. Basal-like vs. HER2-enriched vs. Luminal for Table S3). Therefore, for the same method, results in Tables 4, S2, and S3 were from the evaluation of the *same* set of models that performs prediction of five BRCA subtypes, while different definitions of the labels were used for evaluation across these tables. When considered jointly, Tables 4, S2, and S3 can comprehensively reflect the classification performance while considering the hierarchical relationship among BRCA subtypes.

While block PLSDA, block sPLSDA, and MORONET achieved similar performance on the classification of Normal-like vs. Non-normal-like subtypes in Table S2, MORONET outperformed other methods under the 4-category and 5-category definitions of BRCA subtypes in Table S3 and Table 4. Moreover, block PLSDA and block sPLSDA obtained significantly worse performance than MORONET when distinguishing different subtypes within the Non-normal-like category in Table S3 and Table 4, with macro-averaged F1 score lower than 0.5. Therefore, when considering the hierarchical relationship among different BRCA subtypes comprehensively, MORONET yielded better classification performance than both block PLSDA and block sPLSDA.

We have revised our manuscript to clarify the evaluation approach of BRCA subtypes in the Results section (Lines 153-162).

Following up on comment 5: *Their current way of solving the problem is adding a rule for the number of pre-selected features (the first principal component of the data after pre-processing should explain less than 50% of the variance). Personally, I feel that in this way, if there is a large number of inter-correlated features, the highly correlated ones will still be selected -- it is just that a larger number of features will be selected in total. A more natural way is to prune/clump the features before feature-selection.*

Response: We thank the reviewer for this insightful comment. To demonstrate that MORONET can handle potential redundant features and is robust to different pre-selected features, we performed an additional experiment by applying the minimum Redundancy Maximum Relevance (mRMR) feature selection [1] on the BRCA dataset for training MORONET. We compared the results of MORONET trained with our ANOVA F-value feature selection and MORONET trained with mRMR feature selection, and the results were shown in Table R1. The mRMR feature selection method has been widely used in bioinformatics to remove redundant and correlated features. It uses an incremental selection scheme that selects a feature that minimizes the redundancy and maximizes the relevance at each step, and it has been shown to lead to good classification accuracy [1]. Specifically, for mRMR feature selection, the same number of features was selected in each omics data type using the training data as the ANOVA F-value feature selection shown in Table 1.

From Table R1, we observed that MORONET achieved similar performance with features pre-selected through ANOVA F-value and mRMR. This demonstrates that MORONET is robust to different feature selection methods and could handle potential redundant features in the dataset.

Table R1. Classification results of MORONET on BRCA dataset with the different feature selection methods

Labels	Feature selection method	Evaluation metrics		
		ACC	F1_weighted	F1_macro
Normal-like, Basal-like, HER2-enriched, Luminal A, Luminal B		ACC	F1_weighted	F1_macro
	mRMR	0.824±0.016	0.819±0.021	0.773±0.023
	ANOVA F-value	0.829±0.018	0.825±0.016	0.774±0.017
Normal-like, Non-normal-like		ACC	F1	AUC
	mRMR	0.869±0.010	0.927±0.006	0.925±0.015
	ANOVA F-value	0.869±0.010	0.927±0.005	0.933±0.015
Normal-like, Basal-like, HER2-enriched, Luminal		ACC	F1_weighted	F1_macro
	mRMR	0.853±0.013	0.835±0.027	0.738±0.046
	ANOVA F-value	0.859±0.018	0.845±0.023	0.747±0.032

Reference

1. Peng, H., Long, F., & Ding, C. (2005). Feature selection based on mutual information criteria of max-dependency, max-relevance, and min-redundancy. *IEEE Transactions on pattern analysis and machine intelligence*, 27(8), 1226-1238.

Reviewers' Comments:

Reviewer #3:

Remarks to the Author:

The authors have addressed the reviewer's comments.